# Rapid Determination and Quality Control of Pharmacological Volatiles of Turmeric (*Curcuma longa* L.) by Fast Gas Chromatography–Surface Acoustic Wave Sensor

**DOI:** 10.3390/molecules26195797

**Published:** 2021-09-24

**Authors:** Yanyan Lu, Jianbo Wang, Gang Shen, Jiuling Liu, Hongwei Zhu, Junning Zhao, Shitang He

**Affiliations:** 1Institute of Acoustics, Chinese Academy of Sciences, Beijing 100190, China; luyanyan@mail.ioa.ac.cn (Y.L.); liujiuling@mail.ioa.ac.cn (J.L.); zhuhongwei1211@mail.ioa.ac.cn (H.Z.); 2Sichuan Academy of Chinese Medicine Sciences, Chengdu 610041, China; yyswjb@yeah.net (J.W.); lyy10991069@163.com (G.S.)

**Keywords:** gas chromatography–surface acoustic wave sensor, gas chromatography-mass spectrometry, turmeric, pharmacological volatiles, determination, quality control

## Abstract

Introduction: A novel analytical method using fast gas chromatography combined with surface acoustic wave sensor (GC-SAW) was developed for rapid determination of the pharmacological volatiles of turmeric (*Curcuma longa* L.). Methods: The volatile compounds in 20 turmeric samples, collected from different parts and different origins, were assessed by the fast GC-SAW. In addition, gas chromatography–mass spectrometry (GC-MS) was employed to confirm the chemical composition of the main volatiles. The digital fingerprint of turmeric was established and analysed by principal component analysis and cluster analysis. Results: Curcumene (9.1%), β-sesquiphellandrene (5.1%) and ar-turmerone (69.63%) were confirmed as the main pharmacological volatiles of turmeric. The content of ar-turmerone in lateral rhizome turmeric was significantly higher than that of top rhizome and ungrouped turmeric. The contents of curcumene and β-sesquiphellandrene in top rhizome turmeric were higher than those in lateral and ungrouped turmeric. The 20 turmeric samples were divided into four categories, which reflected the quality characteristics of the turmeric from different parts and origins. Conclusion: The GC-SAW method can rapidly and accurately detect pharmacologically volatiles of turmeric, and it can be used in the quality control of turmeric.

## 1. Introduction

Turmeric (*Curcuma longa* L.) is a perennial herb, and its dry rhizome is frequently used for medicinal purposes in Asia. It has been found to relieve menstruation pains and amenorrhea, chest and hypochondriacal tingling, and has anti-rheumatic and anti-tumour effects [1,2,3,4]. Turmeric contains curcumin, volatile oils, sugars, sterols, fatty acids, and inter alia, of which curcumin and volatile oils are the main active ingredients. Literature reports have shown that 100 mg/L turmeric volatile oil can shrink the nucleus of lung cancer cells [5]. After adding different concentrations of turmeric volatile oil to A549 lung cancer cells, the number of early and late apoptotic cells increased, indicating that turmeric volatile oil can inhibit the proliferation of cancer cells. Research has shown that the volatile components of turmeric have anticancer, antioxidant, anti-inflammatory, antibacterial, and antiviral effects [5,6]. Therefore, the detection of turmeric active volatiles has a wide range of clinical applications and developmental prospects.

The volatiles of turmeric are generally detected using methods such as high-performance liquid chromatography (HPLC), ultraviolet and visible spectrophotometry (UV), gas chromatography (GC), and mass spectrometry (MS) [7,8,9,10,11]. However, the pretreatment methods for these traditional analysis methods are complex, including steam distillation, solvent extraction, and separation. These processes involve excessive sample manipulation, are time-consuming, have low extraction efficiencies, and can result in the destruction of several volatile compounds. Furthermore, the size of these equipment is too large to meet on-site rapid detection requirements.

The response time of SAW sensors is on the order of microseconds, so the analysis time of the instrument is greatly accelerated. The area of the SAW sensor is only 12 mm^2^, and the reaction speed is fast, so the required length of chromatographic column is greatly reduced. In most SAW sensors, a sensitive coating film is applied to the surface of the SAW sensor. The sensor can then be used to measure several special volatiles. However, the SAW sensors in GC-SAW employ uncoated quartz crystals, and each quartz crystal surface maintains a 500 MHz surface acoustic wave. The sensor maintains good stability and accuracy in a wide temperature range, and has a wider range for testing substances than the coated polymer sensor.

Fast gas chromatography combined with surface acoustic wave sensor (GC-SAW) detection based on the headspace sampling method provides simple, non-destructive, real-time, and rapid detection of a broad spectrum of compounds (volatile and semi-volatile organic compounds), with high sensitivity (pg level) and rapid heating system (10 °C/s) [12,13,14]. The characteristic advantage of the GC-SAW method is its effectiveness as an analytical method in providing on-site measurements without the need for complex pretreatment of the sample. GC-SAW permits high reproducibility and good sensitivity, making it possible to detect medicine volatiles quantitatively. In addition, headspace solid-phase micro-extraction–gas chromatography–mass spectrometry (HS-SPME-GC-MS) was used to accurately confirm the pharmacological volatiles compounds of turmeric. HS-SPME, as a headspace sampling technology based on the sorption of absorbent-coated fused silica fibres, has been introduced to analyse volatile and semivolatile compounds [14,15,16]. HS-SPME has many advantages, the most important of which is the elimination of many of the interferences arising from the sample matrix.

At present, GC-SAW has been applied in many areas, including water quality monitoring, environment monitoring, and the detection of flowers and medicinal plants [17,18,19,20]. However, it has not yet been used for the analysis of pharmacologically volatile compounds in different parts of turmeric, or for different origins of turmeric. In this paper, the application of GC-SAW in the detection of the pharmacological volatiles of turmeric was studied, providing a new and fast detection technology for quality identification and quality control of turmeric.

## 2. Results

### 2.1. Retention Index Calibration

For the GC-SAW analysis, the retention index was used to characterise the volatile components in turmeric samples, effectively avoiding errors caused by operating factors. The n-alkanes mixed standards were analysed under the same conditions, the retention time of each n-alkane standard peak was recorded, and the Kratz and van den Dool retention index (RI) calculation formula was used to calculate the retention index of each volatile component:$$I_{x}^{T}=100n+100\left(\frac{T_{Rx}-T_{Rn}}{T_{Rn+1}-T_{Rn}}\right)$$

$T_{Rx},T_{Rn}$ and $T_{Rn+1}$ respectively represent the retention temperature of the carbon number x, n and n+1 normal alkane. ($T_{Rn}\;$<$\;T_{Rx}\;$<$\;T_{Rn+1}$)

The C6–C18 standard solution was used to calibrate the instrument, and the chromatogram of the calibration test is shown in Figure 1 and Table 1. The area of each peak is expressed in frequency counts (Hz). The data analysis software of GC-SAW provides a chromatogram showing C6–C18 with retention indices and time. The retention index for turmeric was based on the calibration results of 13 n-alkanes and the retention time of the turmeric, which was directly provided by the analysis software.

### 2.2. Turmeric Ample GC-SAW Test

The profiles of the turmeric volatiles were obtained using fast GC-SAW. The frequency of the SAW sensor was altered when an analyte was adsorbed on the surface of the sensor, which directly affected the detection signal in direct. Figure 2 shows the GC-SAW overlapping chromatograms of the volatile compounds of 20 turmeric samples. Twelve peaks were detected. However, there were eight common characteristic peaks in 20 turmeric samples. The retention indices of the eight characteristic peaks were 1186, 1414, 1457, 1484, 1525, 1572, 1667, and 1773, respectively. Three characteristic peak areas, at retention indices of 1484, 1525, and 1667, were relatively high. Notably, the components found in top rhizome turmeric, lateral rhizome turmeric and ungrouped turmeric from different origins were the same, with the only difference being the proportion of these components.

### 2.3. Turmeric Sample GC-MS Test

The GC-MS total ion flow chromatogram of turmeric is shown in Figure 3. Chemical components of turmeric were detected by GC-MS, and 40 compounds were identified with a matching degree of over 80%. A total of 15 compounds with a peak area percentage greater than 0.25% were analysed. In this study, the SPME-GC-MS could detect 40 compounds in the turmeric, far more than were detected by GC-SAW (12 compounds). The difference in the trapping time required by these two methods to set the proper conditions causes this disparity. The SPME fibre was exposed to the headspace above the sample vial for 30 min in SPME-GC-MS, whereas in GC-SAW, the headspace vapour of the sample was swept into the trap inside the system for only 1 s.

The information of the components of GC-MS and GC-SAW is shown in Table 2. Each volatile compound detected by fast GC-SAW was identified by a comparison with GC-MS analysis results. The compounds with the highest content in mass spectrometry were curcumene (22.78%), zingiberene (7.5%), β-stilbene (5.86%), β-sesquiphellandrene (15.89%), and ar-turmerone (18.83%), accounting for 70.9% of the total volatile components of turmeric. The compounds with a higher content in GC-SAW were curcumene (9.1%), β-sesquiphellandrene (5.1%), and ar-turmerone (69.63%). The peak area of ar-turmerone was higher in GC-SAW because the SAW sensor was more sensitive to substances with higher boiling point and higher molecular weight. 

Ar-turmerone, turmerone, and curlone have similar chemical structures, and are hence difficult to separate by GC-SAW but can be clearly distinguished by mass spectrometry fragment ion peaks in GC-MS. As shown in Figure 4 of the mass spectrum, ar-turmerone contains a benzene ring and methylheptenone, which are de-electronised to form C_15_H_20_O^+^ (*m*/*z* = 216). Demethylation (-CH_3_) produces C_14_H_17_O ^+^ (*m*/*z* = 201). Further cleavage leads to higher abundance of C_5_H_7_O^+^ (*m*/*z* = 83) and C_9_H_11_^+^ (*m*/*z* = 119). This is because aromatic compounds can easily stabilise a positive charge by delocalisation.

Turmerone is synthesised by the fusion of a cyclohexadiene and methylheptenone. Turmerone was demethylated (-CH_3_) to form C_14_H_19_O^+^ (*m*/*z* = 203) (Figure 5), with further cleavage to the higher abundance form C_5_H_7_O^+^ (*m*/*z* = 83).

Curlone belongs to the sesquiterpene compounds (Figure 6). After dehydrogenating (-H) and demethylation (-CH_3_), C_14_H_17_O^+^ (*m*/*z* = 201) was generated and was further decomposed to C_9_H_12_^+^ (*m*/*z* = 120) with high abundance. 

According to the MS data, the compounds of seven characteristic peaks detected by GC-SAW were identified. They are as follows: the peak at the retention indices of 1186, 1414, 1457, 1484, 1525, 1572, and 1667 were attributed to α-terpineol, caryophyllene, trans-α-bergamotene, curcumene, β-sesquiphellandrene, curcumol, and ar-turmerone, respectively. However, the retention index of peak 1773 did not find the corresponding chemical substance.

Monoterpene and sesquiterpenes compounds are the main volatile components of turmeric. Monoterpene compounds such as α-terpineol has antibacterial effects. The sesquiterpenes such as caryophyllene, trans-α-bergamotene, curcumene, β-sesquiphellandrene, and curcumol have anti-inflammatory effects. The main component ar-turmerone has anti-inflammatory, anti-malarial, anti-cancer, and neuroprotective effects [21,22,23]. The literature has shown that ar-turmerone can protect the dopaminergic neurons in the midbrain, as evidenced by slice cultures of their activities [22]. Ar-turmerone has inhibitory effects against plasmodium falciparum 3D7 [23]. 

In the analysis of different parts of turmeric from the same origin, the ar-turmerone content of lateral rhizome turmeric was considerably higher than that of the top rhizome and ungrouped turmeric, and the curcumene and β-sesquiphellandrene contents in top rhizome turmeric were considerably higher than those in lateral rhizome and ungrouped turmeric from Yunnan (Figure 7). The same is true of the turmeric in the Lotus Pond markets 1, 2, and 3, Zhagu Town, Tielu, and Muchuan County. Pharmacological activity can be determined by analysing the difference in the volatile components of the different turmeric parts.

### 2.4. Principal Component Analysis and Cluster Analysis of the Turmeric Samples 

The eight characteristic peaks of the 20 turmeric samples in GC-SAW were statistically analysed by principal component analysis and cluster analysis using SPSS 25.0. From the characteristic roots and contribution rates of the principal components, the cumulative variance accounted for by the first three principal components amounted to 82.336%. This indicates that the first three principal components can represent the first eight characteristic peaks that were used to analyse the 20 turmeric samples. Therefore, the first three indicators can be extracted. The three principal components are denoted as A, B, and C (first, second and third principal components), respectively. Cluster analysis was used to analyse the three principal components A, B, and C and classify the 20 turmeric samples. According to the contents of the main components, a pedigree diagram and a three-dimensional distribution map for the 20 turmeric samples were formulated. A good classification of different parts and different origins was obtained (Figure 8 and Figure 9).

The 20 turmeric samples were divided into four categories (Table 3). The first category was the lateral rhizome and top rhizome turmeric from Lotus Pond Market 3. The second was the lateral rhizomeand top rhizome of turmeric from Yin-an Village and Lotus Pond Market 1. The third was the top rhizome of turmeric, mainly from Lotus Pond Market 2, Zhagu, Tielu Town, and Lotus Pond Market 1. The fourth was ungrouped turmeric from Myanmar, Yunnan, Yulin, Tielu Town, Xintang, Shilong, and Baijiawan Village. 

## 3. Materials and Methods

### 3.1. Materials

GC-SAW was made by the Institute of Acoustics, Chinese Academy of Sciences in Beijing, China. GC-MS was made by Aglient technology company in American. Thermo-static heater was provided by Xiamen Yudian technology company. Ethanol (≥99.5%, no water level) and N-alkanes (≥99.5%) were provided by Aladdin Biochemical technology limited company in Shanghai, China. Nitrogen (99.999%) was provided by Beijing Zhaoge Gas technology limited company in Beijing, China.

Turmeric samples were provided by the Sichuan Academy of Traditional Chinese Medicine and were identified as genuine by researcher Qingmao Fang of Sichuan Academy of Traditional Chinese Medicine. The turmeric samples were divided into top rhizome, lateral rhizome and ungrouped turmeric according to the different parts of the rhizome. Top rhizome turmeric is the rhizomes of turmeric seeds grown on production. Lateral rhizome turmeric refers to rhizomes growing on the top rhizome [6,24]. Ungrouped turmeric refers to rhizomes where top rhizome or lateral rhizome cannot be identified. Turmeric samples were dried at 60 °C and then powdered, screened, and stored at 4 °C. The origins and parts of 20 turmeric samples are shown in Table 4.

### 3.2. GC-SAW Analytical Conditions 

In this study, fast GC-SAW technology was used for rapid headspace sampling detection of turmeric volatiles from different sources. A schematic diagram of this process is shown in Figure 10 [12]. The GC-SAW is equipped with an injection port, pump, six-port valve, trap, GC column, and SAW sensor. GC-SAW uses two steps process, the sampling and analysis processes. During the sampling processes, the samples entered the trap through the sampling pump. During the analysis processes, the six-port valve was switched, and the trap was heated at high temperature, and the carrier gas was transported from the trap to the GC column. Due to the different separation effects of different compounds in the GC column, the time required for different compounds to reach the detector is different. Qualitative analysis was performed according to the peak time for each compound. Different compound qualities changed the frequency of the detector, allowing quantitative analysis of the compound.

The set-up temperatures were 50 °C for the detector, 200 °C for the injection port, and 160 °C for the valve. The trap was heated rapidly to 200 °C to desorb. The column was heated from 55 °C to 180 °C at a rate of 5 °C/s in 5% phenyl methyl polysiloxane DB-5 (1 m × 0.25 mm). Nitrogen (99.999%) was used as a carrier gas at 4.0 mL/min. Duplicate measurements were carried out per sample. All analytical procedures were completed within 30 s. 

### 3.3. GC-SAW Retention Index Calibration Method and Sample Analysis

The n-alkane mixed standards solution was analysed under the same conditions. The retention time of each n-alkane standard peak was recorded, and the in-house software was used to calculate the retention index according to the retention index calibration formula. For the GC-SAW analysis, the retention index was used to characterise the volatile components in turmeric samples, effectively avoiding errors caused by operating factors. 

After the calibration of retention index, approximately 20 mg of each turmeric powder sample was placed in a 40 mL glass vial sealed with a Teflon–silicone septum. The capped vial was maintained at 30 °C for 2 h to establish equilibrium prior to analysis. The turmeric samples were tested by headspace injection.

### 3.4. Headspace Solid-Phase Micro-Extraction Method

Approximately 0.4 g of turmeric sample was placed in a 40 mL glass vial sealed with a screw cap containing a Teflon–silicone septum. The capped vial was kept at 25 °C to equilibrate for 1 h before HS-SPME sampling. The SPME fibre was exposed to the headspace above the sample vial at 25 °C for 30 min. After adsorption, the SPME fibre was extracted from the sample vial and immediately inserted into the injection port of the GC-MS.

### 3.5. GC-MS Analytical Conditions

Mass spectrometry test conditions: the set-up temperature was 230 °C for the ion source, the scanning mode was 35–350 U, and the electron impact energy was 70 eV. Helium was used as a carrier gas at 1.0 mL/min.

Gas chromatography test conditions: The RTX-5MS column (30 m × 0.25 mm × 0.25 µm) was maintained at 50 °C for 5 min, heated from 50 °C to 250 °C at a rate of 5 °C/s, and maintained at 250 °C for another 5 min. Helium was used as a carrier gas. The set-up temperature was 250 °C for the inlet port and 250 °C for the interface.

## 4. Conclusions

On the basis of this study, we conclude that the fast GC-SAW analytical method shows good sensitivity at the picogram level, rapid detection ability within tens of seconds, and high reproducibility, thereby facilitating the detection and quantification of pharmacologically volatiles of medicinal materials. The simplicity of GC-SAW can reduce the damage caused by multi-step pretreatments in traditional methods. Recognisable active volatile analysis also facilitates distinguishing plant part and origin. The GC-SAW method can serve as an alternative analytical technology for the analysis of active volatiles in turmeric. In addition, to our knowledge, this is the first study to evaluate the pharmacological volatiles in turmeric using GC-SAW.

## Figures and Tables

**Figure 1 molecules-26-05797-f001:**
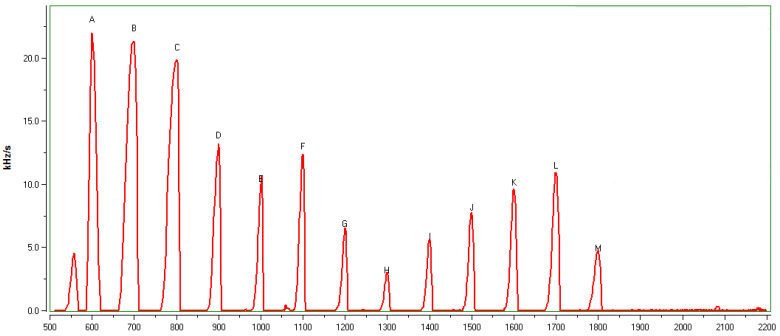
Calibration test chromatogram of 13 n-alkanes standard solution.

**Figure 2 molecules-26-05797-f002:**
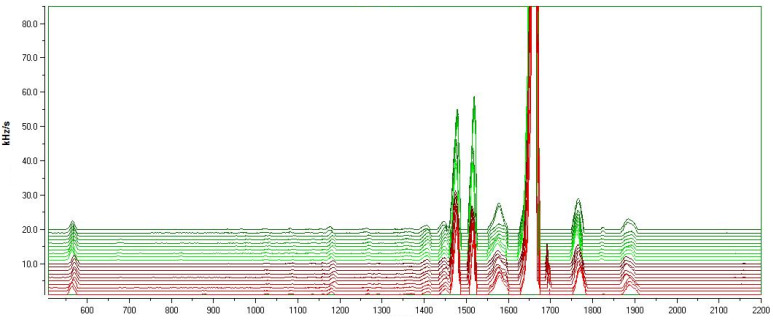
GC-SAW chromatograms of 20 turmeric samples (from top to bottom are turmeric samples numbers 1–20).

**Figure 3 molecules-26-05797-f003:**
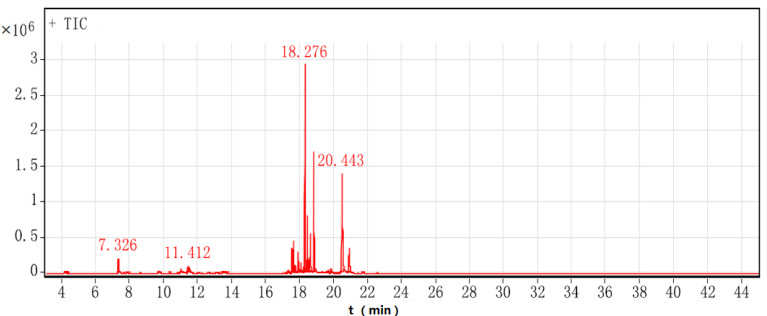
GC-MS total ion chromatogram of turmeric.

**Figure 4 molecules-26-05797-f004:**
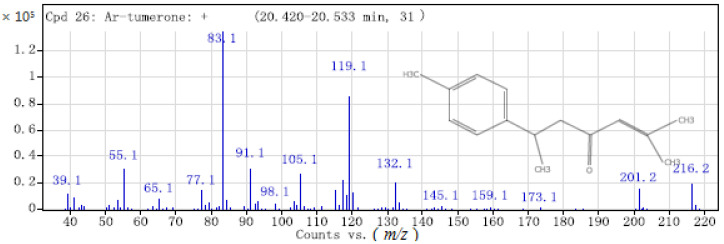
Mass spectrum of ar-turmerone.

**Figure 5 molecules-26-05797-f005:**
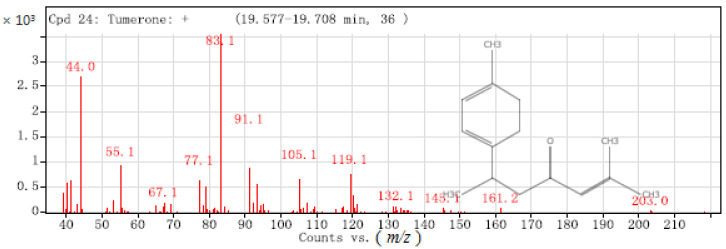
Mass spectrum of turmerone.

**Figure 6 molecules-26-05797-f006:**
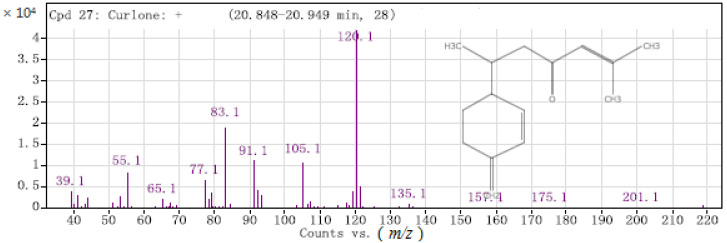
Mass spectrum of curlone.

**Figure 7 molecules-26-05797-f007:**
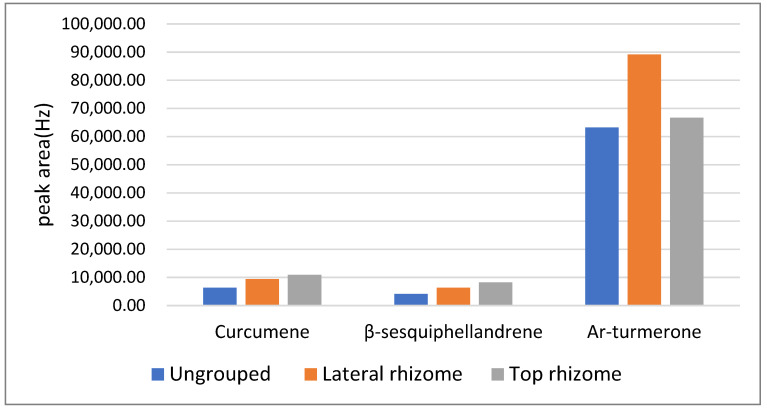
Comparative map of curcumene, β-sesquiphellandrene and ar-turmerone in different parts of turmeric in Yunan by GC-SAW.

**Figure 8 molecules-26-05797-f008:**
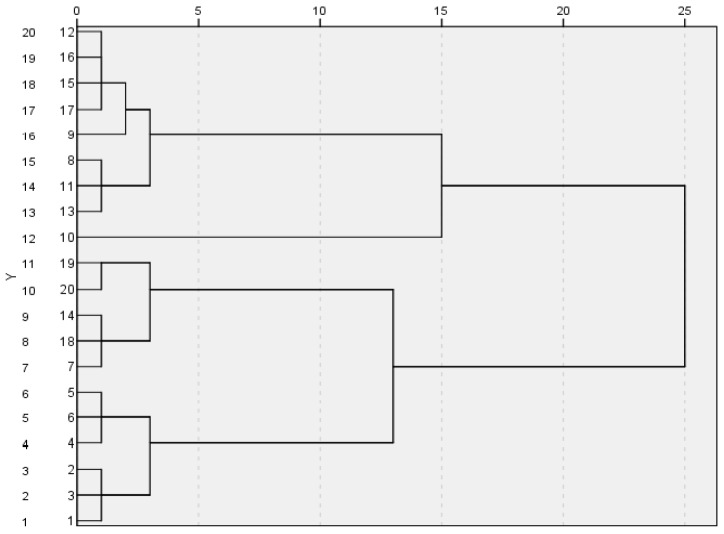
Cluster analysis pedigree of 20 turmeric samples.

**Figure 9 molecules-26-05797-f009:**
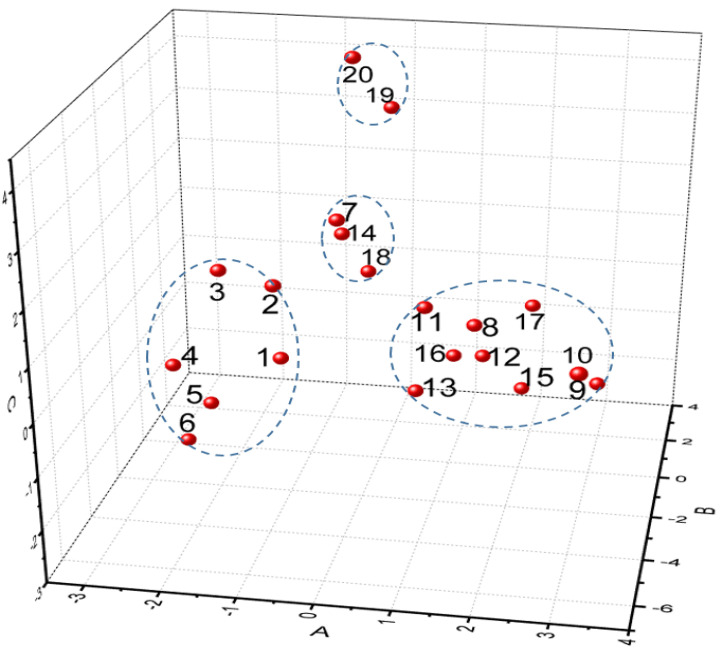
Three-dimensional distribution map of 20 turmeric samples.

**Figure 10 molecules-26-05797-f010:**
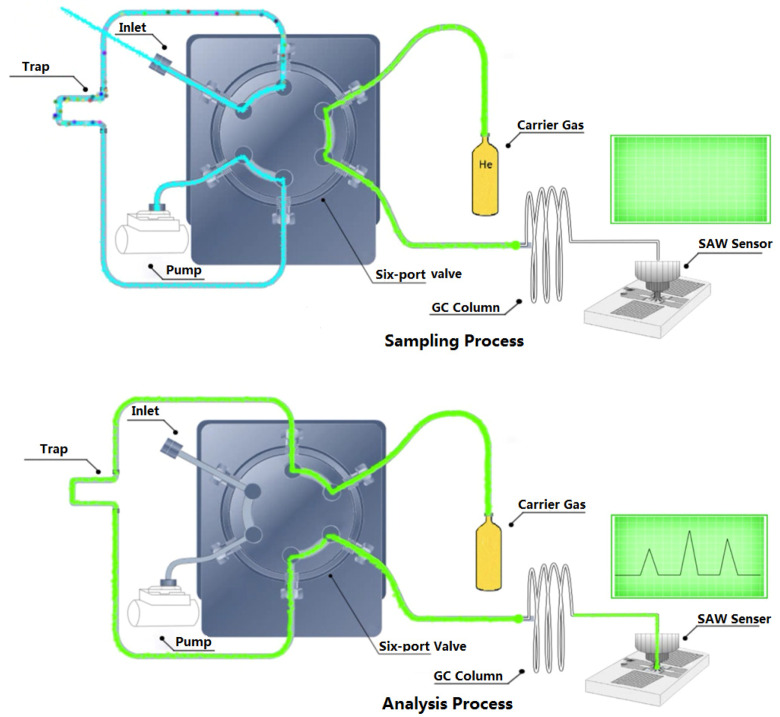
Schematic diagram of GC-SAW.

**Table 1 molecules-26-05797-t001:** Calibration table of 13 n-alkanes standard solution.

No.	Time(s)	Retention Index	Pea Area
A	0.56	600	1800
B	0.94	700	2603
C	1.62	800	3789
D	2.56	900	2311
E	3.90	1000	1563
F	5.76	1100	3151
G	7.84	1200	1746
H	10.20	1300	842
I	12.60	1400	1640
J	14.86	1500	2273
K	16.84	1600	2740
L	18.58	1700	3051
M	20.08	1800	1314

**Table 2 molecules-26-05797-t002:** GC-MS and GC-SAW test data comparison of turmeric.

No.	Name	Molecular Formula	GC-MS Retention Time (min)	GC-MS Relative Content (%)	GC-SAW Relative Content (%)	Retention Index	MS Similarity (%)
1	Ethanone, 1-(3-ethylcyclobutyl)	C_8_H_14_O	7.326	3.15	-	-	84.58
2	2-Propyltetrahydropyran	C_8_H_16_O	9.711	0.72	-	911	88.02
3	Furan, 2-butyltetrahydro-	C_8_H_16_O	10.333	0.25	-	981	88.38
4	α-Phellandrene	C_10_H_16_	10.996	1.05	-	1011	92.82
5	Eucalyptol	C_10_H_18_O	11.514	1.04	-	1053	80.84
a	α-Terpineol	C_10_H_18_O	-	-	0.12	1186	-
6	Caryophyllene	C_15_H_24_	17.598	3.24	0.26	1414	98.26
7	Trans-α-Bergamotene	C_15_H_24_	17.703	1.51	0.54	1457	89.67
8	Humulene	C_15_H_24_	18.033	1.76	-	1450	90.76
9	Curcumene	C_15_H_22_	18.276	22.78	9.10	1484	97.36
10	(-)-Zingiberene	C_15_H_24_	18.422	7.5	-	1484	92
11	β-Bisabolene	C_15_H_24_	18.595	5.86	-	1509	95.32
12	β-sesquiphellandrene	C_15_H_24_	18.797	15.89	5.10	1525	96.94
b	Curcumol	C_15_H_24_O_2_	-	-	1.03	1572	-
13	Turmerone	C_15_H_22_O	19.663	0.44	69.63	1632	72.77
14	ar-turmerone	C_15_H_20_O	20.443	18.83		1667	90.99
15	Curlone	C_15_H_22_O	20.87	4.61		1660	91.6

**Table 3 molecules-26-05797-t003:** Classification table of 20 groups of turmeric samples.

Category	Code	Origin of the Collected Turmeric Parts
First category	20,19	Lateral rhizome turmeric in Lotus Pond Market 3 Top rhizome turmeric in Lotus Pond Market 3
Second category	7,14,18,	Lateral rhizome turmeric in Yin-an Village, Jianban Town Lateral rhizome turmeric in Yin-an Village, Jianban Town Lateral rhizome turmeric in Lotus Pond Market 1
Third category	17,16,15,11,12, 13,9,10,8,	Lateral rhizome turmeric in Lotus Pond Market 2 Top rhizome turmeric in Lotus Pond Market 2 Top rhizome turmeric in Lotus Pond Market 1 Lateral rhizome turmeric in Yunnan Top rhizome turmeric in Yunnan Top rhizome turmeric in Tielu Town New Countryside Top rhizome turmeric in Zhagu Town, Muchuan County Top rhizome turmeric in Zhagu Town, Muchuan County Ungrouped turmeric in Zhagu Town, Muchuan County
Fourth category	5,6,4,2,3,1	Ungrouped turmeric in Tielu Town, Muchuan county Ungrouped turmeric in Myanmar Ungrouped turmeric in Xintang village, Xinmin Town Ungrouped turmeric in Guangxi Yulin Ungrouped turmeric in Yunnan Ungrouped turmeric in Shilong Village, Longchi Town Ungrouped turmeric in Baijiawan Village, Longchi Town

**Table 4 molecules-26-05797-t004:** Origin and parts of 20 turmeric samples.

No.	Origins	Parts
1	Turmeric in Tielu Town, Muchuan county	Ungrouped
2	Turmeric in Myanmar	Ungrouped
3	Turmeric in Xintang village, Xinmin Town	Ungrouped
4	Turmeric in Guangxi Yulin	Ungrouped
5	Turmeric in Yunnan	Ungrouped
6	Turmeric in Shilong Village, Longchi Town	Ungrouped
7	Turmeric in Yin-an Village, Jianban Town	Lateral rhizome
8	Turmeric in Zhagu Town, Muchuan County	Ungrouped
9	Turmeric in Zhagu Town, Muchuan County	Top rhizome
10	Turmeric in Zhagu Town, Muchuan County	Top rhizome
11	Turmeric in Yunnan	Lateral rhizome
12	Turmeric in Yunnan	Top rhizome
13	Turmeric in Tielu Town New Countryside	Top rhizome
14	Turmeric in Yin-an Village, Jianban Town	Lateral rhizome
15	Turmeric in Lotus Pond Market 1	Top rhizome
16	Turmeric in Lotus Pond Market 2	Top rhizome
17	Turmeric in Lotus Pond Market 2	Lateral rhizome
18	Turmeric in Lotus Pond Market 1	Lateral rhizome
19	Turmeric in Lotus Pond Market 3	Top rhizome
20	Turmeric in Lotus Pond Market 3	Lateral rhizome

## Data Availability

Data available in a publicly accessible repository.

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
