# Peer review of "Rapid Determination and Quality Control of Pharmacological Volatiles of Turmeric (Curcuma longa L.) by Fast Gas Chromatography–Surface Acoustic Wave Sensor"

_molecules, 2021, doi:10.3390/molecules26195797_

Round 1

Reviewer 1 Report

Dear authors

The paper provide interesting results, which certainly will be of interest to future readers.

However, some statements need to be better justified Incl.:

*you basically use a SAW sensor instead of a more conventional detection method, at the output of your GC column. Why should this method provide a faster way or an easier way to perform the analysis? SAW gas sensors are not faster than conventional detection methods (e.g. QCM, optical...). Please improve this part (i.e. your solution vs. competing/existing solutions)

*What kind of SAW sensor did you use? What about the sensitive layer? Did you use a specific layer, which is more sensitive to the compounds you want to measure? Please elaborate on this, in the paper. What about selectivity? How does this compare to other GC-SAW devices (literature)?

In general, the paper presents many results, but too few technical elements to allow a proper evaluation of the proposed solution. It shall be improved, prior to publication.

The paper is well-written, but contains numerous typos: "(CA). method.", "vfolatiles", "anti-atherosclerosis", "the discriminate of samples", "detection to of volatiles", etc... Please correct.

Author Response

Thank you very much ! Those section have been improved in the text.

Reviewer 2 Report

The manuscript shows the quantification and quality control of volatiles of turmeric by Fast Gas Chromatography-Surface Acoustic Wave Sensor. The manuscript sounds interesting, though I would suggest overall it can be improved a bit in terms of the use of the English language.  

Author Response

(The authors gave the same response as above.)
